# The Impact of Music Therapy in a Pediatric Oncology Setting: An Italian Observational Network Study

**DOI:** 10.3390/healthcare12111071

**Published:** 2024-05-24

**Authors:** Barbara Zanchi, Timothy Trevor-Briscoe, Pierfrancesco Sarti, Veronica Rivi, Lorenzo Bernini, Jenny Burnazzi, Pio Enrico Ricci Bitti, Alessandra Abbado, Elena Rostagno, Andrea Pession, Johanna M. C. Blom, Dorella Scarponi

**Affiliations:** 1Conservatorio Bruno Maderna, 47521 Cesena, Italy; b.zanchi@consmaderna.org (B.Z.); timothy.trevor@unibo.it (T.T.-B.); pioenrico.riccibitti@unibo.it (P.E.R.B.); 2Fondazione Policlinico Sant’ Orsola, 40138 Bologna, Italy; berninilorenzo1@gmail.com (L.B.); burnazzi.jenny@iiscervia.it (J.B.); alessandra.abbado@gmail.com (A.A.); 3MusicSpace Italy Association, 40122 Bologna, Italy; 4Department of Biomedical, Metabolic, and Neural Sciences, University of Modena and Reggio Emilia, 41125 Modena, Italy; pierfrancesco.sarti@unimore.it (P.S.); veronica.rivi@unimore.it (V.R.); 5Department of Psychology, University of Bologna, 40127 Bologna, Italy; 6Oncoematologia Pediatrica, IRCCS Azienda Ospedaliero Universitaria di Bologna, 40138 Bologna, Italy; elena.rostagno@aosp.bo.it; 7Department of Medical and Surgical Sciences, University of Bologna, 40126 Bologna, Italy; andrea.pession@unibo.it; 8Centre for Neuroscience and Neurotechnology, University of Modena and Reggio Emilia, 41125 Modena, Italy; 9IRCSS AOU Sant’ Orsola, 40138 Bologna, Italy; dorella.scarponi@aosp.bo.it

**Keywords:** MT, agency, coping strategies, relationship with parents, network analysis, caregivers, taking the initiative

## Abstract

Background: Music Therapy (MT) is a non-pharmacological, art-based intervention that employs music experiences within a therapeutic alliance to attend to clients’ physical, emotional, cognitive, and social requirements. This is the first study aiming at investigating the impact of MT on the psychological facets of children suffering from cancer. Methods: The study, combining the AQR and m-YPAS assessment tools, evaluated behavioral, sound–musical, and interactive parameters in pediatric oncology patients undergoing MT sessions during hospitalization. Fifty patients admitted to the Paediatric Oncology and Haematology Unit at Policlinico S. Orsola Hospital in Bologna, Italy, were enrolled, irrespective of their treatment regimen. Data collection occurred on the first day of the MT session between 3 p.m. and 5 p.m., with observations conducted by independent observers. In addition to traditional statistical analysis, network analysis was used to explore the combined interactions of all parameters, effectively discerning the distinctive roles played by each one during therapy sessions and their influence on all others. Results: Network analysis highlighted distinct patterns of interactions among parameters during the various sessions, emphasizing the role of positive emotions and a calm setting, the child’s ability to take the initiative in sessions, their sense of agency, and the parent’s role in guiding them. Significant differences were recorded at each time point between all variables considered. Conclusions: The results of this innovative study may pave the way for future multicenter studies aimed at further exploring the role of MT in children undergoing both curative and palliative treatments for cancer.

## 1. Introduction

A cancer diagnosis has significant physical and psychological consequences and presents an existential challenge for patients, their families, and the healthcare professionals involved [1,2,3,4,5]. This is especially true for pediatric cancers whose diagnosis at a young age often evokes profound emotional and psychological responses, potentially resulting in the development of trauma and psychosomatic symptoms [6]. These experiences may persist into adulthood and contribute to the onset of psychiatric disorders later in life [6]. 

Over the last two decades, new trends and holistic approaches have been included in the oncology setting [1,4]. Among them, Music Therapy (MT) represents a non-pharmacological, art–based approach that uses music experiences within a therapeutic relationship to address clients’ physical, emotional, cognitive, and social needs [7]. An essential element of all types of MT intervention is the sound–music synergy, which aims at enhancing the expressive abilities and resources of each patient [2,3,5]. This approach has been shown to have a profound impact on patients’ well-being, offering relief from physical and psychological symptoms [8,9]. In particular, its positive effects on patient anxiety, stress, mood, perceived quality of life, and sense of isolation [1,9], make it particularly suitable for use in pediatric oncology [3]. Importantly, the interventions can be tailored to the patient’s needs to facilitate emotional expression, communication, and social interaction, allowing them to express their (positive and negative) feelings in a safe and non-judgmental environment [10,11,12] and interact with peers going through similar experiences, thus promoting social skills useful to contrast the loneliness that accompanies the hospitalization period [13,14,15]. MT not only benefits patients but also offers opportunities for family bonding and emotional support for caregivers [15]. Involving parents and family members in MT sessions helps create a robust network of support for pediatric patients throughout their illness [16]. 

Despite advancements in MT techniques, studies focusing on pediatric oncology are limited, highlighting the need for further research in this area [17]. While innovative methods in MT have been explored for both pediatric and adult patients within multidisciplinary teams, research on MT’s effects in oncology and palliative care has predominantly focused on adults. This underscores the necessity for additional investigation specifically in the pediatric oncology setting. In 2009, MT was introduced to the “Unità Operativa Pediatrica Pession Paediatric Oncology and Haematology, Policlinico S. Orsola Hospital in Bologna” (Italy), thanks to funding from the Mozart 14 Association led by Alessandra Abbado. Since its introduction, MT has been practiced continuously with pediatric oncology patients and their families by a team of professional music therapists from the MusicSpace Italy Association. The therapy follows an interactive-relational approach, adapting to patients’ changing needs. 

The primary aim of this study was to investigate the potential of MT in counteracting emotional withdrawal and disengagement among patients within the same therapy session. This could result in heightened active involvement, diminished anxiety levels, and an increased adoption of positive coping mechanisms. Our study particularly focused on the assessment of behavioral indicators (i.e., arousal, the capability of making choices, eye contact, and facial expressions), sound–musical aspects (i.e., interactions with objects, instruments, and musical elements), and interactive dynamics (both with music therapists and within group settings) within pediatric oncology patients undergoing MT sessions. Changes in the children’s psycho-behavioral parameters were recorded at five different time points during the MT session. As these sessions were carried out in a controlled clinical environment, changes in the indices measured, compared to T0 (pre-assessments in a pre-post intervention study), could primarily be attributed to MT. 

The truly innovative aspect of the study involved the development and testing of a novel assessment framework. This framework combined the “AQR—Assessment of the Quality of Relationships” and the “m-YPAS, Yale Preoperative Anxiety Scale”. This integrated approach enabled the evaluation of diverse psychological parameters, including arousal levels, eye contact, and verbal and physical interactions with parents and medical professionals, in a quicker and more targeted manner, considering the unpredictable behavior of children while maintaining the same level of accuracy. 

To our knowledge, this is the first use of network analysis to analyze parameters collected during MT sessions in a pediatric oncology setting. Together, the new evaluation grid and the methodological approaches used in this study could pave the way for future studies aimed at promoting the use of MT in the onco-pediatric setting. 

## 2. Materials and Methods

### 2.1. Study Design

The study conducted is a prospective observational type in which the effects of the intervention were evaluated by following the young patients from the beginning of the music therapy session (pre-intervention—T0) to its conclusion (post-intervention—T4).

The music therapist team follows an interactive-relational approach based on the clinical sound–music therapy “free improvisation” model. This model was put forward by J. Alvin and is based on the assumption that when environmental conditions promote the freedom to choose and play without any pre-imposition (free improvisation), the person’s characteristics, pathology and problems will be reflected in the music [18]. This model is close to the concept of “free associations” in Freudian psychoanalysis and sees music as a means of projection. Furthermore, this model takes into account both listening to music and actively making music through instruments or vocalizations.

At a pediatric age, joint improvisation between therapist and child allows aspects of interaction and attachment to be identified [19,20].

Due to patient complexities and privacy concerns, sessions could not be filmed, and collected material was later analyzed. The efficacy of the therapy was assessed, focusing on a single session lasting up to 60 min. 

During a session, two independent observers (trained music therapists) could complete a minimum of three and a maximum of five assessments: T0—beginning of the session, T1—15 min, T2—30 min, T3—45 min, and T4—60 min. The number of assessments depended on how long the child expressed a willingness to stay within the session. The MT sessions were consistently scheduled in the afternoon, from 3 p.m. to 5 p.m., ensuring uniformity in timing across all sessions. It is pertinent to note that all patients included in the study were receiving medication and were not in the terminal phase of their illness. Within the designated MT room, two music therapists were present to facilitate interactions with the children, offering flexibility in engagement based on individual needs. While one therapist assumed the primary role, the other acted as a co-therapist, with these roles dynamically shifting based on the evolving dynamics of each therapeutic encounter. Importantly, these roles were not predetermined but rather adapted organically to suit the unique requirements of each patient interaction. Simultaneously, two independent observers, both trained music therapists, conducted their observations remotely from separate rooms, ensuring an impartial assessment of each child participating in the study. This approach maintained consistency and objectivity in data collection while allowing for a comprehensive evaluation of the therapeutic process.

Moreover, considering ethical concerns and existing stress, we aimed to evaluate the impact of MT effectively. Since it was not possible to exclude this procedure from the standard protocol for pediatric cancer patients receiving treatment, nor to omit to propose it to parents, it was impractical to attain a uniform and sufficiently large control group. Consequently, in the statistical analysis and discussion of the findings, we designated the T0 phase of the study as the pre-intervention time point, so the assessment was conducted before the onset of the MT sessions. By employing this approach, the results illustrate the alterations within the same session compared to T0.

Importantly, MT adheres to the Standardized Instrument for Pediatric Oncology (SIPO) guidelines, which provide a framework for evaluating interventions and outcomes in pediatric oncology settings. MT aligns with these guidelines by incorporating standardized instruments like the AQR and m-YPAS to assess various parameters, such as behavioral, sound–musical, and interactive aspects, in pediatric oncology patients undergoing therapy sessions. This adherence ensures that MT interventions are aligned with established standards and facilitates a reliable evaluation of their effectiveness in improving patient outcomes within the pediatric oncology context.

### 2.2. Development of a New Evaluation Grid

In our study, we conducted a thorough literature review on MT in pediatric and oncology settings, identifying two scales to assess specific parameters of interest: the “AQR—Assessment of the Quality of Relationships” [21] and the “m-YPAS, Yale Preoperative Anxiety Scale” [22]. These scales cover aspects like arousal, eye contact, choice-making, and interactions with parents and physicians, addressing concerns observed during clinical sessions.

The “AQR-Tool for the Assessment of the Quality of Relationship” was developed by music therapist Schumacher K. and psychologist Calvet C., primarily for children with an autism spectrum disorder [21]. It evaluates the patient–therapist relationship across developmental stages using four scales: instrumental engagement, vocal-pre-speech engagement, physical–emotional engagement, and therapist intervention appropriateness. The “m-YPAS” is an observational checklist measuring preoperative anxiety and assessing behaviors in hospital and operating room settings. While we could not use these scales in their original form due to limitations (like no video recording), we adapted their concepts to create an observation grid specific to MT at St. Orsola Hospital. This customized tool evaluates real-time parameters: behavior, sound–musical aspects, and interactions. The observation grid (Table 1) was collaboratively designed by music therapists, clinicians, and researchers and includes eight parameters/items, indicating patient progress during the session. Parameters one to seven are rated on a five-point Likert scale (0 = low, 4 = high), while the eighth and most specific parameter to MT, “Use of Music in the Relationship” (Table 2), uses an eight-point scale to assess the child’s engagement with music. This parameter is based on the AQR scale. The two tools with their respective a priori parameters were combined in one single observation group.

To finalize the evaluation grid, a 6-month pilot study was conducted to train independent music therapist observers. During the first month, training took place with the two observers, followed by 20 observation sessions, one per week. Independent observations made by the two observers on eleven patients were recorded: ten were undergoing treatment and one had just been diagnosed and was about to start treatment. All observations were supervised by a senior music therapist to define and apply the grid in the best way possible. From these observations, an acceptable level of inter-reliability was achieved, which measured 93% between the two observers.

To ensure the integrity of the subscales within the original questionnaires, items from various scales assessing the same construct were not utilized. This was to prevent any potential multicollinearity between the scores of the scales and to preserve the internal validity of the subscales.

### 2.3. Participants

A total of 76 children (mean age 6.6 ± 4.3) were recruited for this study (43 females, mean age 6.6, and 33 males, mean age 6.5). All these children had a diagnosis of onco-hematological disease. To be considered eligible for the study, the young patients needed to

-Have a diagnosis of an onco-hematological disease;-Be undergoing treatment in the pediatric onco-hematological ward of S. Orsola Hospital;-Have conducted the music therapy session not in their own room but in the space of the ward designated for this intervention;-Have remained inside the room in which music therapy was carried out for at least 30 min (thus having collected at least three assessments of the child);-Have received informed consent to the research from the parents/legal guardians of the hospitalized children.

All patients who did not meet the inclusion criteria were therefore excluded but, of course, MT continued to be included as part of their care. Out of 76, only 50 (21 males and 29 females) were considered eligible. A family member was always present during the sessions, most often mother and father but in some cases also grandparents or siblings. 9 out of 50 children had no relatives present in the room together with the music therapists.

All parents of the participants were fully informed about the study, its methods, data processing, and the possibility of publishing the data in such a way as to maintain the anonymity of the patients involved. They agreed to participate and signed the Informed Consent to Participation and Data Processing for publication.

### 2.4. Statistical and Network Analysis

First, data were analyzed for the assumption of normality using the Kolmogorov–Smirnov one-sample test for normality (K-S distance and P) and the Shapiro–Wilk test for normality. Preliminary analyses were then carried out to test the homogeneity of variances between groups and independence using Levene’s test. We processed the scores of the eight scales of the observational grid (i.e., arousal, eye contact, positive facial expression, making choices, taking the initiative, vocalization/verbalization, and relationship with parents) and compared them with those from each time point. 

The Friedman one-way repeated-measures analysis of variance by ranks (Friedman’s test) was used to analyze the results between T0, T1, T2, T3, and T4. The post hoc analysis was conducted by comparing the results of two possible options after using Friedman’s test using the Conover test for multiple comparisons of mean rank sums and the Wilcoxon signed-rank test. All tests were defined as significant at *p* < 0.05. Data were presented as mean ± standard error.

A first level of stratification was performed by analyzing the performance of male and female patient scores separately to check for significant differences between the various time points. All statistical analyses were performed using SPSS v. 28.0 (IBM Corp., Armonk, NY, USA) and R software (version 4.0.3/2020-10-10), while graphs were generated using GraphPad Prism v. 9.00e for Windows^®^ (GraphPad Software, Inc., La Jolla, CA, USA). 

To understand the importance of each variable and consider all the other parameters, five network models [23] were constructed to analyze the partial correlations between the variables and, using the centrality indices [24], identify which were the most influential variables at each time point that mediated part of the variation of the others. The network models represented were implemented using R software (version 4.0.3/10 October 2020). The package “psychometrics” [25] was chosen to calculate the network models. This package allows for the creation of a Gaussian Graphical Model (GGM) [26] based on the input data, which is then extracted and represented using the “graph” package [27]. The GGM forms an undirected network model in which edges represent partial correlation coefficients. A Gaussian Graphical Model (GGM) visually depicts the conditional relationships among variables using a graph. In this graph, each node represents a variable, while the edges between nodes signify conditional dependencies or nonzero partial correlation coefficients. By generating a network that graphically represents the interaction of the variables considered, the centrality measures of “Strength” and “Betweenness” were extracted to define and understand which node (variable), at each time of the study, was the one with the greatest importance in the network. The initial metric denotes the quantity of links a node possesses with others. In a weighted network such as the ones in this article, it is determined by multiplying the number of nodes connected to a given node by the average weight of these connections, modified by a tuning parameter. The second measure assesses the extent to which a node participates in the shortest routes between other nodes. It helps identify nodes that are probable connectors between other nodes, thus indicating which nodes are most likely to facilitate connections within the network. More information on these indices can be found elsewhere [16,28].

## 3. Results

The analyzed sample of children with haemato-oncological disease consisted of 50 patients (mean age 6.6 ± 4.3): 21 males (mean age 6.5 ± 3.1) and 29 females (mean age 6.6 ± 4.9). For all parameters observed by two independent trained observers, there was an increase in the score from the beginning of the MT sessions to detection after 60 min. The only two parameters to decrease in the transition from T3 to T4 (between 45 and 60 min) were “Arousal”, dropping from a score of 3.40 to 2.70 (highly significant), and “Relationship with Parents” from 3.30 to 3.10 (not significant). All mean scores for each survey time are shown in Table 3. Friedman’s test indicated multiple significance for all variables taken into consideration by the evaluation grid when the sample of 50 children was analyzed. No significant differences were identified between males and females after comparing the scores at each time point. As the pattern of scores between the two sexes is not the same for each parameter investigated, this requires more attention with a larger sample of subjects. 

As shown in Figure 1, Friedman’s test shows a gradual increase in most parameters, which, compared with T0, become increasingly significant. All variables except “Arousal”, “Positive facial expressions” and “Eye contact” have scores that become significantly different from T0 around 45 and 60 min (T3 and T4). 

Although not significant, the separate analysis of the mean scale scores for male and female patients (Table 3) showed that girls have higher scores in arousal levels (both initial and final), taking the initiative, making choices, and showing positive facial expressions. Boys, although initially seeking less of a relationship with their parents, end up looking for more at T3; they verbally articulate more during the session and use/manipulate musical instruments more toward the end of therapy.

Figure 2 shows the network models that have been generated using the “psychometrics” package. The centrality values returned by the models show that the most important variable in the network (the one with the highest betweenness and strength values) is different at each time point, except for T3 and T4, where the most important characteristic is the change in sign of the edges that connect node 3 (relationship with parents) with the nodes; this indicates an evolution over time in emotional involvement. At time T0, the most important variable is node 6 (positive facial expression) with a betweenness of 1; at time T1, the most important variable is node 5 (making choices) with a value of 1, followed by node 4 (taking the initiative); and at T2, the most important variables are nodes 7 and 8 (eye contact and vocalization) with respective values of 1 and 0.28. At time T3, the most important variable is node 3 (relationship with parents) with a value of 1, and finally, at time T4, the most important variables are nodes 7 and 2 (relationship with parents and arousal), which have values of 1 and 0.71. 

No significant changes in connection density or a decrease in the number of edges were identified. All networks are highly dense (edge density = 1) and are therefore highly plastic and prone to change following external perturbations (both positive and negative). This aspect underlines how the model of the MT session setting, shown in Figure 3, resulting from the interpretation of network analysis models, is extremely valuable. 

The networks demonstrated a method for structuring MT sessions that gradually prioritized aspects directly linked to the child (such as initiative-taking, positive facial expressions, and eye contact) as the session progressed. Additionally, attention was given to relational aspects, including interactions with parents and hospital staff, as well as utilizing music to express oneself and foster connections.

More importantly, the sessions showed a critical point after 30 min, when 26 out of 76 children (34%) were no longer able to continue. 

The network analysis showed how, at this time point and then in the following sessions, in the children who continued, it was possible to integrate intrinsic (vocalizations and eye contact) and extrinsic (active search for the presence of the parent) relational aspects.

## 4. Discussion

In this exploratory study, we used a novel evaluation grid by merging the “AQR—Assessment of the Quality of Relationships” and “m-YPAS, Yale Preoperative Anxiety Scale” to assess crucial parameters like arousal, eye contact, and interaction with parents and physicians. This innovative grid demonstrated high adaptability, reliability, and ease of administration, enabling the efficient collection of vital psychological and behavioral data, meticulously examined through statistical and network analysis. To our knowledge, this is the first time employing such an approach to evaluate MT’s impact on the mental well-being of young patients within pediatric oncology.

Moreover, the network analysis managed to track the dynamic evolution of variables throughout music therapy sessions, representing an additional innovative approach to the study. By visualizing the progressive succession of variables and their interactions, therapists and clinicians can gain invaluable insights into the emotional growth of their clients. This method not only enhances comprehension but also offers a more nuanced understanding of the therapeutic process, potentially leading to more effective interventions and outcomes.

Overall, the results indicate a positive trajectory across all evaluated parameters during MT sessions. The observed increase in these parameters strongly aligns with the existing literature, correlating “eye contact”, “positive facial expressions”, “decision-making”, “initiative-taking”, “vocalization”, and “parental interaction” with enhanced “active engagement” strategies and positive control over their environment, fostering emotional self-regulation abilities [13,29,30,31]. 

Our analysis revealed a single decrease in the scores of arousal and the relationship with parents during the transition between T3 and T4, possibly due to patient fatigue or heightened calmness. For patients in this age group, engaging in therapies that extend beyond highly interactive sessions can become very demanding. Notably, substantial score improvements were primarily observed between T0 and T4, with significant differences across other intervals. As sessions progressed, children experienced benefits in re-establishing a sense of “normality”, manifested as heightened eye contact, improved decision-making, positive emotions, and proactive behavior linked to a regained sense of agency and self-efficacy. These gains occurred despite an initial perception of an unpredictable and uncertain environment—a critical consideration within pediatric oncology where patient vulnerability is an everyday reality [32,33,34]. This is particularly relevant in a pediatric oncology setting, as the capacity and opportunity to exercise influence over their surroundings hold immense significance for these young patients given the daily perception of vulnerability due to their physical conditions and the uncertainty that accompanies their future [13,14,35,36].

Network analysis enabled the identification of specific roles of variables at each time point, allowing for the delineation of a pattern of children’s behavior throughout the MT sessions (see Figure 3). This pattern can be utilized to optimize outcomes during the sessions.

-T0: Since it is not possible to actively intervene, in an identical and controlled manner, before the sessions begin, it is important to set aside time to put the child at ease and in a positive state of mind to promote their activation and interest in the activity.-T1: The child’s resourcefulness becomes evident as they realize they are in a space where they can exert control, deciding how to behave and interact with the objects around them. At this point, it becomes essential to encourage their initiative and allow them to make choices independently.-T2: At 30 min, the phase begins in which the child seeks a relationship with others, particularly through eye contact and vocalizations. The child seeks involvement and judgment from others (adults) in the room, in particular from the parents/parents present.-Critical moment: Between T2 and T3, there is a transitional phase where relational behavior “markers” shift from being intrinsic to extrinsic. The child actively looks for the caregiver, and the relationship with the caregiver becomes central at T3. The parent plays a pivotal role.-T4: If the “crisis” is resolved positively, the network at T4 illustrates that the node representing the relationship with parents has changed the direction of all connections previously negative, with nodes like “Arousal”, “Making choices”, and “Vocalization,” and strengthened some of those already established. This underscores how the relationship with the parent/caregiver is essential in enabling the child to complete 60 min of the session and achieve the maximum increase in scores on most of the evaluated scales.

There are two main limitations of this study: the first limitation is the absence of a control group that underwent a different activity during the sessions compared to MT. Unfortunately, this was not feasible, as already mentioned in the “Methods” section, due to guidelines on treating pediatric oncology patients. T0, the initial assessment before the session began, was used as the control group. Statistical analyses were chosen taking this aspect into account. 

The second limitation concerns the use of a multivariate non-parametric repeated-measures test to simultaneously assess the effect of the time points and patients’ gender (given the differences in means). The test used was the Scheirer–Ray–Hare test (a non-parametric test used for a two-way factorial design). Due to its limited use and the choice not to perform individual non-parametric t-tests between time points considering only “gender” as a comparison, the authors preferred to omit the discussion of ‘patterns’ in the presence of non-significant differences. 

## 5. Conclusions

The results of this innovative study approach lay the groundwork for future multi-centric investigations and analyzing the differences while stratifying for gender and, with adequately sized samples, other variables such as the presence of parents or other relatives, tumor type, and aspects related to children’s resilience. Overall, the aim was to study the role of MT by addressing the multifaceted needs of children undergoing cancer treatment, encompassing physical, emotional, social, and developmental dimensions, both in curative and palliative contexts.

Our new approach allowed for an exploration of MT’s impact on the psychological facets of children facing cancer. A comprehensive literature review was followed by the identification of two assessment scales (AQR and m-YPAS) to evaluate parameter changes during MT sessions. Rigorous statistical analyses demonstrated positive trends, further elucidated through network analysis. 

This strategic approach unveiled the intricate interplay among all eight parameters, illuminating the distinct “roles” each parameter assumed during the sessions. Furthermore, this method facilitated the exploration of centrality indices, guiding the trajectory of the MT sessions.

Future research should prioritize expanding the scope of variables investigated to the fullest extent possible, particularly given the high-risk nature of this patient population. This will enable a more comprehensive understanding of the interplay between socio-cultural, demographic, and economic factors in shaping various behavioral and resilience profiles. Additionally, it will be crucial to explore the long-term effects of cycles of music therapy sessions on young cancer patients to ascertain whether this approach could offer broader applicability in oncology as an alternative therapy.

## Figures and Tables

**Figure 1 healthcare-12-01071-f001:**
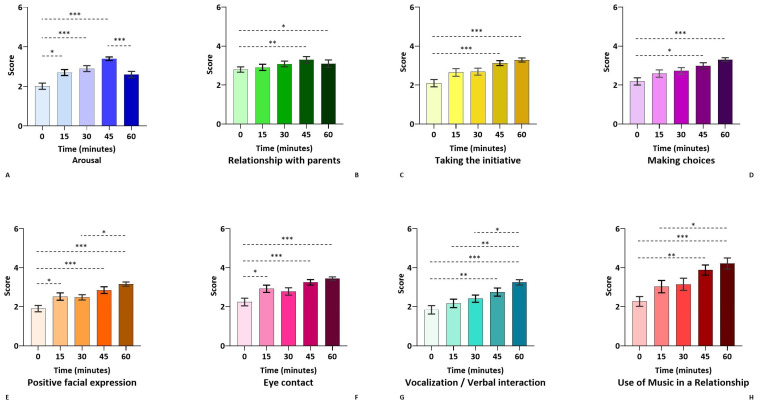
Effects of the MT section on different parameters, measured at different time points: at the beginning of the session (T0), 15 min (T1), 30 min (T2), 45 min (T3), and 60 min (T4) later. Data have been analyzed using non-parametric repeated-measures ANOVA of each variable entered into the evaluation grid. *p*-values are listed when significant: *** *p* < 0.001, ** *p* < 0.01; * *p* < 0.05. Subfigures (**A**–**H**) represent the eight parameters separately. The colour coding of each parameter is subsequently reflected in the nodes of Figure 2. (**A**): Arousal; (**B**): Relationship with parents; (**C**): Taking the initiative; (**D**): Making choices; (**E**): Positive facial expression; (**F**): Eye contact; (**G**): Vocalization/verbal interaction; (**H**): Use of music in the relationship.

**Figure 2 healthcare-12-01071-f002:**
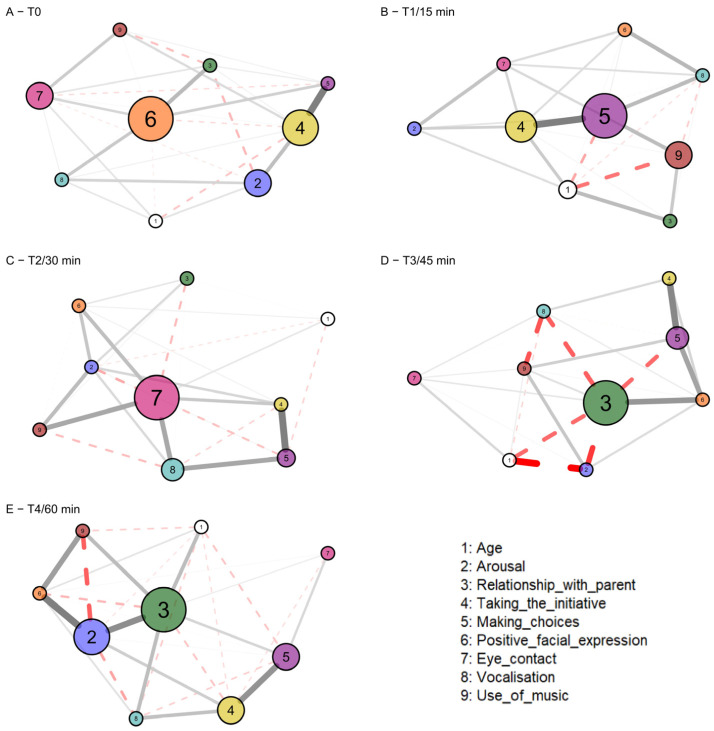
Representation of the five network models, one for each detection time. The nodes represent the evaluation grid variables and the variable “Age”. The node colors reflect those in Figure 1. The links between the nodes (edges) indicate the presence of a partial correlation between the two variables. The thicker the line, the greater the correlation value. Black edges indicate a positive partial correlation while red dashed edges indicate a negative one. The size of each node represents its value of betweenness centrality: the bigger the node, the higher the value.

**Figure 3 healthcare-12-01071-f003:**
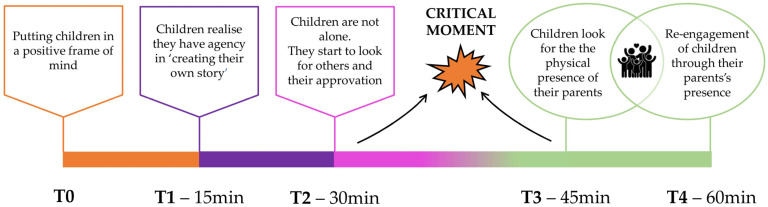
Summary of the evolution of MT sessions through the application of network models. The colors of the boxes and circles match the nodes in Figure 2 with the highest values of betweenness centrality.

**Table 1 healthcare-12-01071-t001:** Description of each one of the observational grid parameters scored by the music therapists at each time point.

(1) **Arousal.** Refers to how alert the child is. Measured by observing how the child looks around and how they react to the music therapist’s actions and sounds. The scale ranges from a minimum value—when the patient’s body is withdrawn and contracted, when they avert their gaze or when the gaze is absent, or if they wince about sounds while in the setting—to a maximum value—when the patient’s body is relaxed, their gaze attentive, and they have a pleasant reaction to sounds.
(2) **Eye contact.** Refers to the frequency with which the child makes eye contact with the music therapist, with other children, or with other persons involved during the session. The scale ranges from a minimum value indicating total absence/avoidance of eye contact to a maximum value corresponding to the presence of eye contact with no avoidance of gaze.
(3) **Positive facial expression.** Refers to facial expressions demonstrating positive emotions and affects. The minimum value indicates a total absence of facial relaxation and smiles, a prevalence of expressions indicating emotions such as anger, sadness, fear, or disgust; the maximum value indicates a marked frequency over time of expressions showing happiness and/or serenity.
(4) **Making choices.** Refers to the participant’s ability to make choices. The choice is defined as the indication of a preference manifested verbally or through a gesture. The scale ranges from a minimum value indicating no choices to a maximum value indicating making many choices.
(5) **Taking the initiative.** Refers to when the participant initiates a conversation or addresses a question to another person without this being a response to a request or indication. It also includes indicating an object or a person or initiating a musical/sound conversation. The scale ranges from a minimum value indicating the absence of such behavior to a maximum value indicating a strong presence.
(6) **Vocalization, verbal interaction.** Refers to the participant’s use of their voice. The scale ranges from a minimum value that indicates the absence of verbal response, even when crying, to a maximum value that indicates the child’s engagement in producing sound (or in actively listening if s/he does not or cannot speak while taking part in an activity such as playing a musical instrument), asking questions, or making comments. If the child is too young to speak, then laughter, babbling, or sound production in a calm state are associated with a maximum value.
(7) **Relationship with parents.** Refers to the child’s behavior towards their parent. The scale ranges from a minimum value when an anxious, insecure, and ambivalent type of attachment is observed, to a maximum value for a secure attachment that allows the child to engage in play and age-appropriate behavior without the need to turn to the parent, with whom he/she may nevertheless interact if the interaction is initiated.
(8) **Use of music in a relationship.** The use of music in a relationship is measured through eight different levels/modes of relationship that the patient establishes with the musical instruments, their sounds, and those produced by the music therapist. These modes range from no contact at all to a musical and verbal relationship. As with the previous seven parameters, the observer also notes down every 15 min which of the eight modes best defines the use of music in a relationship for each child.

**Table 2 healthcare-12-01071-t002:** Possible evaluations of the observer concerning parameter 8, “Use of Music in the Relationship”, of the observational grid.

(0) “**No contact**”: Patient’s total disinterest in the instruments in the room; no contact or relationship with them.
(1) “**Minimal contact**”: The patient develops an initial contact with the instruments. This occurs briefly after making a sound by chance. The instrument is touched and then totally disregarded.
(2) “**Functional sensorial contact**”: The patient makes sensory, destructive, or stereotyped use of the instrument. Sensorial use is defined as the instrument being explored through touch, taste, or smell instead of through sound. Destructive use is defined as if the instrument is in danger of being damaged. A monotonous, unchanging, and meaningless way of playing is read as stereotyped.
(3) “**Sense of self—use of the instrument as one’s own**”: The patient explores the instrument by recognizing it as a musical instrument and treats it with appropriate effect.
(4) “**Contact with others**”: The patient plays the instrument appropriately and the resulting sound is related to the sounds produced by the music therapist.
(5) “**Interaction**”: The patient plays the instrument in the context of a dialog, as in a question-and-answer game; often, instrumental production is associated with verbal expressions.
(6) “**Shared experience-shared affectivity**”: The patient plays the instrument with pleasure and with constantly positive emotions. Playing can also lead to associations. The use of the instrument helps to demonstrate an affective state in a fun way.
(7) “**Verbal musical space**”: The use of the instrument triggers emotional changes and imaginative content, leading to verbalization (reflection/description).

**Table 3 healthcare-12-01071-t003:** Means of each variable entered in the evaluation grid at each time point. The maximum values reached are shown in bold. T0 = start of session, T1 = 15 min, T2 = 30 min, T3 = 45 min, T4 = 60 min. For each variable, the separate values of males and females have also been reported.

	This Means at Each Time Point				
PARAMETERS	T0	T1	T2	T3	T4
Males	1.83	2.69	2.71	3.36	2.31
Arousal	2.01	2.70	2.90	**3.40**	2.60
Females	2.14	2.71	3.03	3.43	2.81
Males	2.71	3.17	3.02	3.52	3.14
Relationship with Parents	2.80	2.91	3.09	**3.30**	3.10
Females	2.86	2.72	3.14	3.14	3.07
Males	2.07	2.38	2.67	2.88	3.05
Taking the Initiative	2.10	2.65	2.69	3.13	**3.29**
Females	2.12	2.84	2.71	3.31	3.47
Males	2.02	2.31	2.81	2.60	3.24
Making Choices	2.19	2.59	2.73	2.99	**3.31**
Females	2.31	2.79	2.67	3.28	3.36
Males	1.88	2.05	2.45	2.48	2.88
Positive Facial Expressions	1.90	2.52	2.48	2.85	**3.16**
Females	1.91	2.86	2.50	3.12	3.36
Males	2.29	2.64	2.69	3.31	3.52
Eye Contact	2.24	2.92	2.78	3.25	**3.44**
Females	2.21	3.12	2.84	3.21	3.38
Males	1.95	1.95	2.43	2.36	3.36
Vocalization/Verbalization	1.84	2.17	2.41	2.75	**3.25**
Females	1.76	2.33	2.40	3.03	3.17
Males	2.17	3.02	2.88	3.45	4.24
Use of Music	2.27	3.03	3.15	3.88	**4.22**
Females	2.36	3.03	3.34	4.19	4.21

## Data Availability

The possibility of sharing raw data is only considered following a formal request to the corresponding author (Johanna M. C. Blom).

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
