# Peer review of "The Impact of Music Therapy in a Pediatric Oncology Setting: An Italian Observational Network Study"

_healthcare, 2024, doi:10.3390/healthcare12111071_

Round 1

Reviewer 1 Report

Comments and Suggestions for Authors

Comments and Suggestions for the Authors:

We felt as though the introduction and citations in the study were robust and clear. The paper did meet the objective of investigating the impact of music therapy using a novel assessment framework and Network Analysis to analyze parameters gathered during therapy sessions. We had no research design concerns.

In the  description, further clarification is needed regarding the statement: "The music therapist team follows an interactive-relational approach based on the clinical sound-music therapy 'free improvisation' model." Consistency in the font size of text in each graph, particularly in Figure 1 and spacing of arrows in Table 1.

Additionally, it would be beneficial to provide an explicit definition of "Betweenness" rather than relying on a citation. While no significant differences were found between males and females, there were observed trends in variables between the sexes. It may be worthwhile to either omit this mention or explore the significance of these trends further.

Future directions could involve considering multiple sessions per patient to evaluate long-term effects and incorporating demographic information about participants, such as socioeconomic status and cultural background.

Comments on the Quality of English Language

--

Author Response

We would like to thank Reviewer 1 for the thorough revision of our manuscript. We appreciate the time and effort that she/he dedicated to providing feedback on our paper and are grateful for the insightful comments on and valuable improvements to our paper.

Reviewer 2 Report

Comments and Suggestions for Authors

This paper is interesting and potentially very interesting but presents some critical issues which, in my opinion, require major revision. 

Here some suggestions:

In the abstract it should be clear what type of study is being done, the title indicates this is an observational study, not in the abstract. It is also not clear which kind of research design has been used. Were the scales administered at what time? Before, after, during a TM session? At another time? At what poi t durino the treatment? Once or several Times? What hypotheses are talked about in the conclusions of the abstract? I suggest we be more clear.

Introduction

A (3) appears on line 68, what does it mean? Is it a mistake?

The introduction is definitely too long and feels a bit heavy for the reader. Some parts can be reduced, others eliminated.

Some suggestions:

What is written by line 75 to line 83 is negligible and could be eliminated. From line 84 to 88 reduced. What is expressed in Lines 89 to 97 is superfluous and can be summarized in the description of the therapeutic approach.

The purpose of the study is correctly expressed in the introduction but must be summarized.

What is expressed from line 107 to line 122 must be condensed in the method and materials section and in the conclusions section.

2.1 Study design

This section needs to be completely redone.

There is no description of the type of research was used and an accurate description of the research design must to be added. 

A specific section should be dedicated to describing the type of MT approach used.

line 130; reference is made to "psychological parameters collected before, during and after the sessions", which parameters are you talking about? How were they collected?

Is the study based on 5 assessments done over a single session?

The impossibility of having a control group has been well explained and it is important to underline that MT compare within the SIPO guidelines.

The measurements made at T0 cannot be considered as a "control group" but as pre-assessments in a pre-post intervention study.

The article does not describe what type of music therapy is done and if it is done, at what moment of the treatment/day, by one or more people, this represents an important lack.

2.2

The development of an observation grid derived from the AQR would deserve to be described in a separate publication. It is not explained how the Inter-rater reliability was assessed, how many observers were involved, in short, how the pilot study was conducted.

2.3

It would be very useful to make a table with the characteristics of the participants.

Why were 3 measurements taken for some sessions (so were multiple sessions evaluated? If so, how many?) and for others 5?

What does this difference depend on?

It is necessary to explain what the inclusion and exclusion criteria were.

It is not clear whether the observers who filled out the forms are blinded, whether they know the aims of the study, how "independent" they are, what type of training they have and the possible biases they might have. (A music therapist, for example, might have an interest in overestimating what he observes, a MD in underestimating what he observes).

It is necessary to understand whether the observation grid is compiled using “a priori” parameters, or not. In conclusion, the tool should be better described.

Results

The use of network analysis is very interesting and the identification of a critical moment within the session/sessions is very interesting.

Discussion

From line 318 to line 323 the authors report the failure to achieve a result which however was not among the objectives of the study. 

Author Response

We would like to thank Reviewer 2 for the dedication of time and effort in providing feedback on our paper, and we sincerely appreciate the insightful observations and valuable enhancements she/he has contributed to our work.
